# Unlocking regioselective *meta*-alkylation with epoxides and oxetanes via dynamic kinetic catalyst control

Peng-Bo Bai[1], Alastair Durie[2], Gang-Wei Wang [1] ✉ & Igor Larrosa [2] ✉

Regioselective arene C−H bond alkylation is a powerful tool in synthetic chemistry, yet subject to many challenges. Herein, we report the *meta*-C−H bond alkylation of aromatics bearing *N*-directing groups using (hetero)aromatic epoxides as alkylating agents. This method results in complete regioselectivity on both the arene as well as the epoxide coupling partners, cleaving exclusively the benzylic C−O bond. Oxetanes, which are normally unreactive, also participate as alkylating reagents under the reaction conditions. Our mechanistic studies reveal an unexpected reversible epoxide ring opening process undergoing catalyst-controlled regioselection, as key for the observed high regioselectivities.

Chelation-assisted transition-metal-catalyzed $sp^2$ C−H bond alkylation has emerged as a straightforward approach to forge the fundamentally important $C(sp^2)$−$C(sp^3)$ bond[1–3]. This transformation provides a precisely-controlled complementary approach to the classic Friedel-Crafts alkylation, and has accelerated the rapid assembly of structurally diverse natural products, drug scaffolds and other functional molecules from simple starting materials[4,5]. Alkylating reagents, such as alkyl halides, alkylboron, alkylzinc, alkyltin, and Grignard reagents as well as alkenes have been extensively utilized in C−H alkylation reactions[2]. While less explored, recently epoxides have received increasing attention as alkylating reagents (Fig. 1A) displaying a number of advantages[6,7]: 1) epoxides are stable and readily available; 2) alkylation is redox-neutral and the product incorporates all atoms from the electrophile, thus increasing atom economy; and 3) the resulting hydroxide group in the product is a useful functional handle for further diversification.

In 2015, Kanai and Kuninobu reported a seminal palladium-catalyzed aromatic *ortho*-C−H alkylation with terminal alkyl-epoxides[8]. Concurrently, the Yu group reported the *ortho*-C−H alkylation of benzoic acid with aliphatic epoxides to produce a variety of 3,4-dihydroisocoumarins[9]. Later, Hirano and Miura[10] and Kuninobu[11] extended such transformations by using Ni and Mn catalysis. Meanwhile, the Dong[12,13] and Zhou[14,15] groups reported the use of epoxides as coupling partners with aryl halides in Catellani-type processes,

whereas Wang and others[16–18] also significantly expanded the utility and scope of epoxide opening triggered C−H alkylation reactions (Fig. 1A). However, several key aspects have not yet been explored: firstly, in all examples to date alkylation is exclusively occurring at the *ortho*-C−H bond, with *meta*- and *para*-C−H bond alkylations proving elusive; Secondly, terminal or symmetric 1,2-dialkyl-substituted epoxides are predominantly used with ring opening invariably occurring at the less substituted carbon. Thirdly, the use of oxetanes or other homologs of epoxides in C−H alkylation remains a prominent challenge.

In contrast to directed $sp^2$ *ortho*-C−H bond functionalization, transition-metal-catalyzed *meta*-C−H bond functionalization is significantly more challenging and far less developed[19–24]. Chelation-assisted strategies using either template-directing groups[25–31], hydrogen-bonding ligands[32–35] or transient norbornene mediators[36–39] have been used to activate and functionalize the *meta*-C−H bond (Fig. 1B, a, b, c). As an alternative approach, Frost, Ackermann and others reported that arene *meta*-C−H functionalization can be achieved by Ru-catalysis[24,40–53], proceeding via initial *ortho*-cycloruthenation followed by functionalization *para* to the ruthenium (Fig. 1B, d). This approach avoids the multi-step syntheses of ligands or templates, and provides a powerful platform for *meta*-alkylation of arenes bearing common nitrogen-based directing groups, with secondary or tertiary alkyl halides predominantly used as alkylating reagents. Inspired by this

[1]State Key Laboratory of Applied Organic Chemistry & College of Chemistry and Chemical Engineering, Lanzhou University, Lanzhou 730000, China. [2]School of Natural Sciences, Department of Chemistry, University of Manchester, Oxford Road, Manchester M13 9PL, United Kingdom. ✉e-mail: wanggw@lzu.edu.cn ; igor.larrosa@manchester.ac.uk

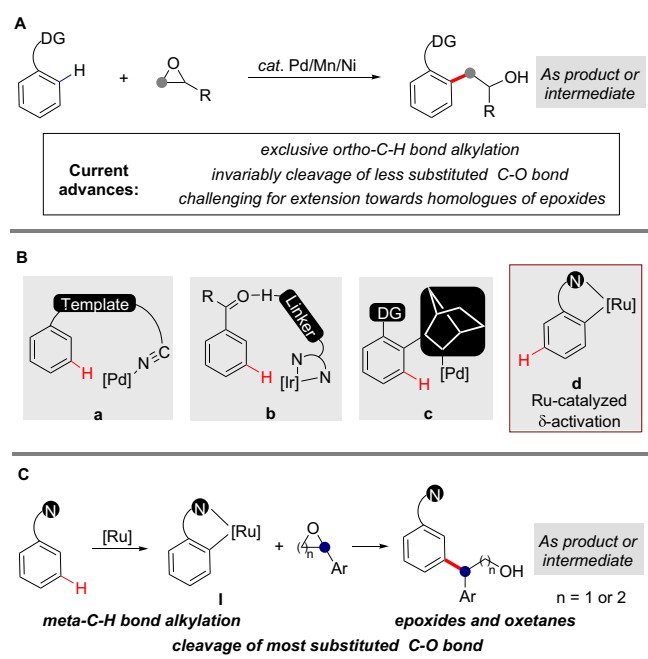

**Fig. 1 | Development of directed C−H alkylation enabled by epoxide and oxetane opening. A** *ortho*-C−H bond alkylation enabled by epoxide-opening (current advances). **B** Strategies for chelation-assisted *meta*-C−H functionalization. **C** Our work: *meta*-C−H bond alkylation with epoxides and oxetanes.

work, we hypothesized that *meta*-alkylation using epoxides as coupling partners may be achievable via Ru-catalysis, thus achieving complementary selectivity to that previously reported with epoxides. Herein, we report our latest work on *meta*-C−H bond alkylation with epoxides (Fig. 1C). This transformation proceeds with complete regioselectivity in both coupling partners, through exclusive cleavage of the more hindered benzylic C−O bond of (hetero)aromatic epoxides and complete *meta*-alkylation selectivity. It is noteworthy that the normally unreactive oxetanes, homologs of epoxides, also participate as alkylating reagents in the reaction. Our mechanistic studies reveal that an unexpected reversible ring opening of epoxide process, as well as a catalyst-controlled discrimination of reaction intermediates, are simultaneously in operation during the reaction, contributing to a dynamic kinetic regioselection, as the key to the high selectivities obtained.

## Results

### Optimization of the reaction conditions

We initially chose 2-phenylpyridine **1a** and styrene oxide **2a** as model substrates (Table 1), which would lead to important pharmacophores of 1,1-diarylalkane unit[54] equipped with a hydroxide group for further product manipulation (**3a**). A key challenge for this design is the requirement to control regioselectivity of both the phenylpyridine and epoxide. For example, a feasible oxidative addition or nucleophilic attack onto the highly electrophilic styrene oxide[6,7], would lead to the undesired *ortho*-alkylation product (**3a′**). When we used [Ru(*p*-cymene)Cl₂]₂ and Ru(*p*-cymene)(OPiv)₂ as catalysts, both of which are predominately used in *meta*-alkylations[40–53], together with benzoic acid (**A1**) as an additive to enhance the reactivity of the epoxide as well as to facilitate the *ortho*-C−H bond cycloruthenation step to form **I**[55], no detectable alkylation product was formed (entries 1-2). The addition of 30% NaI, which has been shown before to facilitate reactions with epoxides[56–58], also led to no reaction (entries 3-4). Gratifyingly, when cyclometalated ruthenium complex **RuBnN**, developed by our group[59–62], was used as the catalyst, the desired product **3a** was

observed for the first time, albeit in low yield and accompanied with trace amount of *ortho*-C−H alkylation product **3a′** (entry 5). We have previously observed that **RuBnN** tends to lead to *ortho*-selective alkylations with secondary alkyl halides[60]. A screening of other reaction parameters resulted in no improvement, with *ortho*-alkylated **3a′** obtained as the major product when using methanol as solvent (entry 6). When commercially available Ru(PPh₃)₃Cl₂ was used, both **3a** and **3a′** were obtained (entry 7). The nature of the acid additive proved to be crucial to the success of this reaction (entries 8-10), with 2-ethylbutanoic acid (**A4**), affording **3a** in 50% yield while completely suppressing the formation of **3a′** (entry 10). The carboxylic acid additive may function as a proton shuttle in the reaction by protonating the alkoxide generated after the epoxide ring opening, with the resulting carboxylate acting as a base to facilitate *ortho*-C−H ruthenation. A further increase in yield was gained when the reaction was run at higher concentrations (1.3 M, entry 11). Although turn-over was observed when 30% of NaI was used, a slightly higher yield of 68% of **3a** was obtained when 1.0 equiv of NaI was used (entry 12), and finally 75% isolated yield was reached when reaction was run at 70 °C instead of 80 °C (entry 13). Meanwhile, similar efficiency was observed using tetrabutylammonium iodide (entry 14) whereas NaBr or NaCl provided significantly lower yields of **3a** (entries 15-16).

### Scope of the reaction

Next, we investigated the reaction scope (Fig. 2). 2-Phenylpyridine derivatives bearing electronically diverse *para*-substituents efficiently participated in the reaction with **2a**, delivering *meta*-C−H alkylation products **3a**-**3i** in excellent yields (Fig. 2A). Functional groups such as phenolic hydroxyl **3b**, alkyl ether **3d**, chloride **3f**, aliphatic ester **3h** and internal alkyne **3i** were well tolerated, leaving room for further product diversification. When 2-phenylpyridine bearing an *ortho*-fluoride substituent was used, the alkylation exclusively occurred at the more hindered *meta*-position (**3j**). This result is consistent with a mechanism involving formation of **I**, followed by a Ru-mediated *para* σ-activation[51]. Some *meta*-substituted phenylpyridines can give product from the reaction but only to limited degree (**3k** and **3l**). In accordance with previous Ru-catalyzed *meta*-alkylation[40–53], *meta*-substitution on the arene is not well tolerated as it forces the cyclometalation to occur on the distal *ortho*-position and then blocks reactivity. This reasoning is also why bis-alkylation does not occur in these reactions. Other heteroarenes, such as pyrimidine, pyrazole and non-aromatic 4,5-oxazoline were also suitable directing groups, producing **3m**-**3o** in moderate to excellent yields. Noteworthy, a tandem reaction was discovered for **1p** bearing an ester group at *para* position, and isochroman-1-one derivate **3p** was formed in 77% yield (Fig. 2B). This tandem lactonization likely occurs via intermediate **II**. This type of tandem processes have been previously observed in *ortho*-alkylations with epoxides where the directing group itself is an electrophile[8–10], while our product **3p** was obtained through such tandem process occurring with a substituent within the aromatic ring. Interestingly, when using **1q** bearing a *para*-aldehyde substituent, identical product **3p** was obtained. This product may be formed through a Ru-mediated β-hydride elimination of **III** (Fig. 2B)[63], or upon work-up in the presence of air. These results clearly highlight the unique utility of this *meta*-alkylation process.

Styrene oxides bearing either electron withdrawing or donating groups at *para*, *meta* or *ortho*-positions also reacted smoothly to yield the corresponding *meta*-alkylation products in moderate to excellent yields (**3q**-**3ad**) (Fig. 2C). Moving beyond styrene oxides, the sterically hindered 1-naphthalene and 1,1′-binaphthalene substituted epoxides were converted into the desired products **3af** and **3ag** in reasonable to good yields. Moreover, electron rich heteroarenes, such as furan and thiophene, or electron deficient quinoline, were also tolerated in the epoxide, leading to products **3ah**-**3aj** in moderate to good yields. Noteworthy, when styrene oxide bearing an *ortho*-ester substituent

**Table 1 | Reaction optimization**

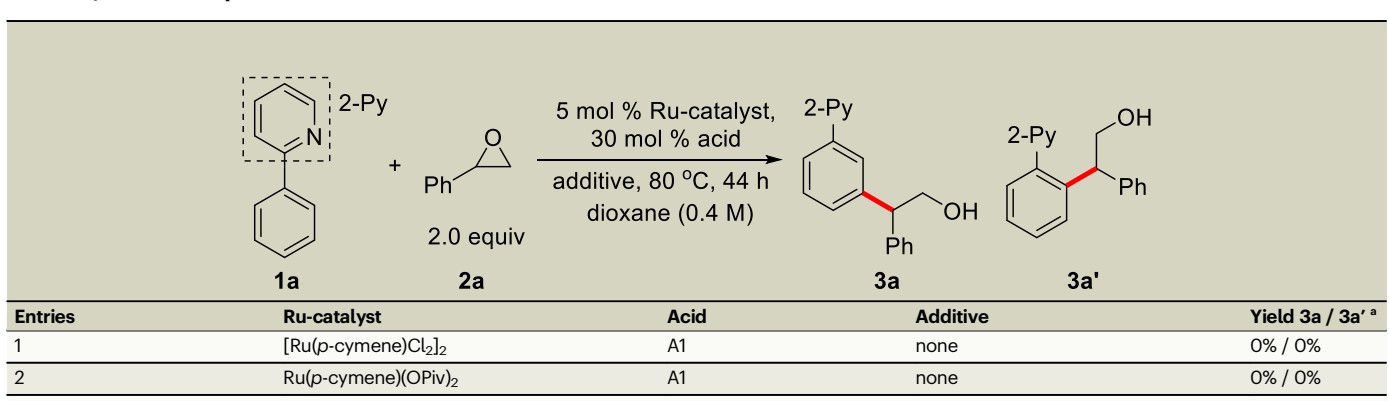

| Entries | Ru-catalyst | Acid | Additive | Yield 3a / 3a' [a] |
|---|---|---|---|---|
| 1 | [Ru(p-cymene)Cl₂]₂ | A1 | none | 0% / 0% |
| 2 | Ru(p-cymene)(OPiv)₂ | A1 | none | 0% / 0% |
| 3 | [Ru(p-cymene)Cl₂]₂ | A1 | NaI (30 %) | 0% / 0% |
| 4 | Ru(p-cymene)(OPiv)₂ | A1 | NaI (30 %) | 0% / 0% |
| 5 | RuBnN | A1 | NaI (30 %) | 27% / trace |
| 6 | RuBnN | A1 | NaI (30 %) | 19% / 25% [b] |
| 7 | Ru(PPh₃)₃Cl₂ | A1 | NaI (30 %) | 15% / 5% |
| 8 | Ru(PPh₃)₃Cl₂ | A2 | NaI (30 %) | 15% / 0% |
| 9 | Ru(PPh₃)₃Cl₂ | A3 | NaI (30 %) | 48% / 0% |
| 10 | Ru(PPh₃)₃Cl₂ | A4 | NaI (30 %) | 50% / 0% |
| 11 | Ru(PPh₃)₃Cl₂ | A4 | NaI (30 %) | 62% / 0% [c] |
| 12 | Ru(PPh₃)₃Cl₂ | A4 | NaI (100 %) | 68% / 0% [c] |
| 13 | Ru(PPh₃)₃Cl₂ | A4 | NaI (100 %) | 75% / 0% [c,d,e] |
| 14 | Ru(PPh₃)₃Cl₂ | A4 | nBu₄NI (100 %) | 73% / 0% [c,d] |
| 15 | Ru(PPh₃)₃Cl₂ | A4 | NaBr (100 %) | 46% / 0% [c,d] |
| 16 | Ru(PPh₃)₃Cl₂ | A4 | NaCl (100 %) | 19% / 0% [c,d] |

[a]Yield measured by NMR using 1,3,5-trimethoxybenzene as internal standard.
[b]MeOH (0.4 M of **1a**) used as the solvent.
[c]Run at 1.3 M concentration of **1a**.
[d]Reaction was run at 70 °C.
[e]Isolated yield.

was used, a similar tandem alkylation-lactonization process was observed, exclusively delivering **3ak**, with an isochroman-1-one unit at the *meta*-position (Fig. 2D).

1,2-Disubstituted epoxides have been shown to be challenging substrates on previous *ortho*-alkylations, leading to regioisomeric mixtures due to the difficulty to differentiate between two C−O bonds with subtle steric and electronic differences. As a result, only a handful of examples of 1,2-disubstituted epoxides have been reported, with most of them being symmetric epoxides[8–18]. When we tested *trans*−2-methyl-3-phenylepoxide as a coupling partner in this reaction (Fig. 2E), the *meta*-C−H alkylation regioisomer **3al** was formed exclusively, resulting from cleavage of the benzylic C−O bond, albeit with a low diastereoselectivity. Identical results were obtained when using *cis*−2-methyl-3-phenylepoxide as the substrate. Similarly, both *trans*-and *cis*−2,3-diphenylepoxide yielded product of **3an** with identical yield and diastereoselectivity. These results are in stark contrast to Miura's work where a stereospecific *ortho*-C−H alkylation is observed[10]. Steric bulk in the epoxide could also be accommodated yielding **3am** in 68% yield.

Finally, indene oxide could be converted into the *meta*-alkylated product **3ao** in 86% yield with complete regio- and excellent diastereoselectivity. These results clearly demonstrate the excellent regioselectivity control of this transformation on both arene and epoxide coupling partners.

The late-stage functionalization of biologically relevant molecules was then carried out to further showcase the utility and tolerance of the transformation (Fig. 2F). Firstly, diazepam was directly reacted with **2a** under standard conditions, with the desired *meta*-alkylation product **3ap** obtained in 82% yield, leaving the amide, imine and chloride functionalities intact. This result also highlights that imines are compatible directing group for the reaction. Benefiting from the easy accessibility of epoxides from acids, aldehyde and alkenes[64], estrone, adapalene and telmisartan were derivatized to include an epoxide motif and then used in the alkylation reaction with **1a**. Pleasingly, the desired complex targets **3aq-3as** were produced in fair to excellent yields.

In an extension of this strategy, the use of oxetanes instead of epoxides was examined (Fig. 3). Despite their potential

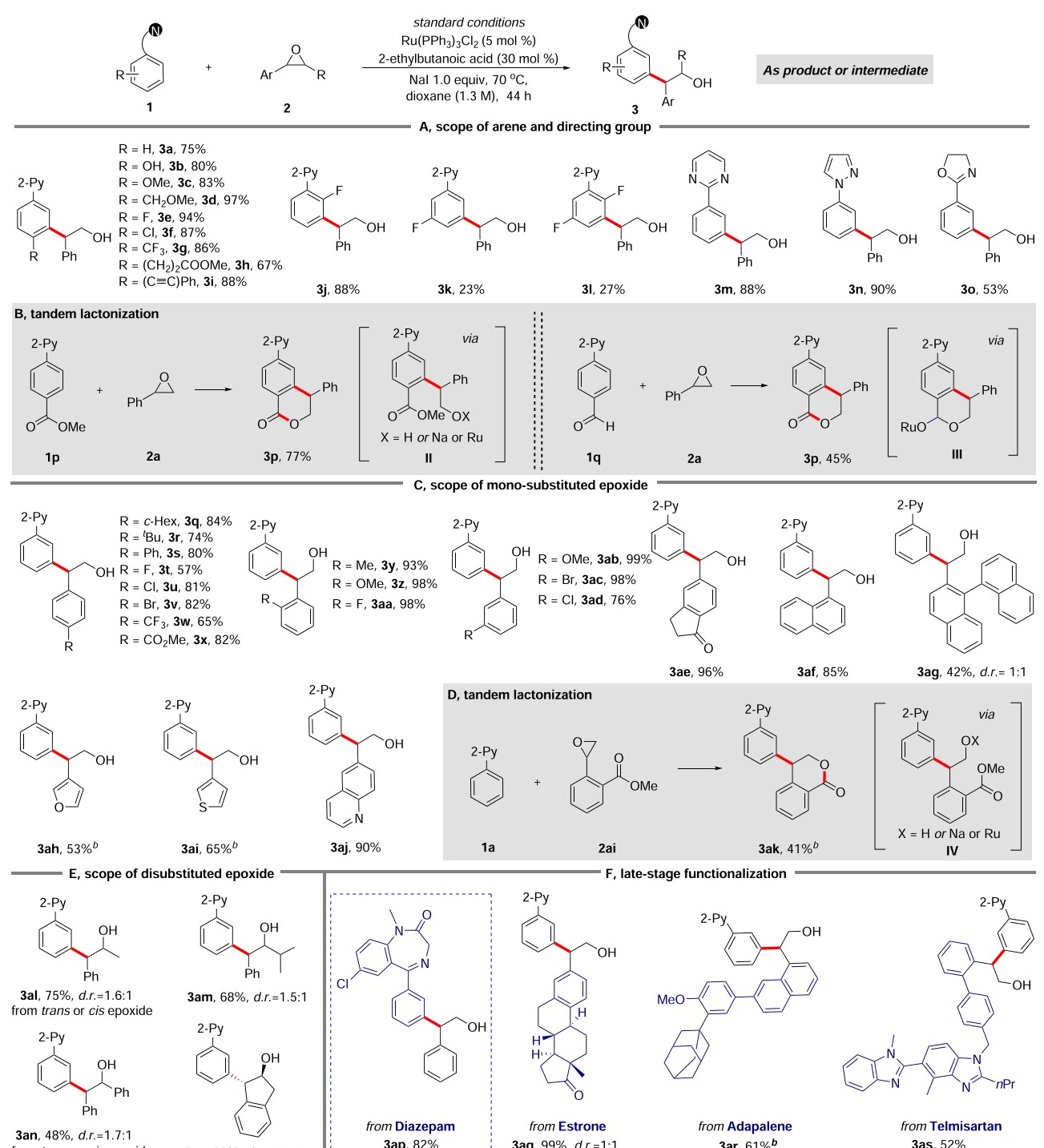

**Fig. 2 | Substrate scope[a]. A** Scope of arene and directing group. **B** Tandem lactonization for arene substrates. **C** Scope of mono-substituted epoxide. **D** Tandem lactonization for epoxide substrare. **E** Scope of disubstituted epoxide. **F** late-stage functionalization of complex moleculars. [a] **1** (0.2 mmol) and **2** (0.4 mmol) were used. Yields are of isolated product; [b] **1** (0.2 mmol), **2** (0.8 mmol), Ru(PPh₃)₃Cl₂ (10 mol %) and NaI (2.0 equiv) were used.

broad versatility as building blocks[65], oxetanes have rarely been used in directed C−H alkylation[66], presumably because of their less strained nature making their activation harder. Indeed, DFT calculations by Fang showed that the free energy barrier of Pd oxidative insertion into oxetane is significantly higher than that of epoxide[67]. We speculated that the iodide additive could lead to nucleophilic opening of oxetane, thus bypassing the high energy oxidative pathway. With a small modification to our standard conditions, 2-phenyloxetane **4** successfully alkylated **1a**, affording the desired *meta*-alkylation product **5a** in 62% yield. Interestingly, a tandem alkylation-lactonization did not occur when **1n** reacted with **4**, likely due to the difficulty of forming a seven-membered lactone, thus yielding **5b** instead in 38% yield. Testing several substituted 2-phenyloxetanes showed that both the electron donating and electron withdrawing substitution could be well tolerated (**5c**-**5g**).

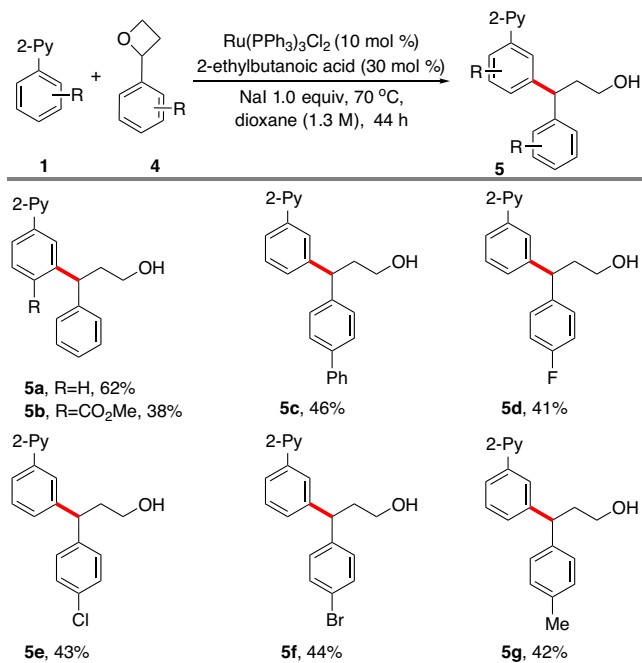

**Fig. 3 | Ring opening of 2-phenyloxetanea.** Yields are of isolated product with **1** (0.2 mmol) and **4** (0.8 mmol) were used.

## Mechanistic studies

A series of experiments were carried out to investigate the reaction mechanism (Fig. 4). First, when TEMPO (2.0 equiv) was used as a radical scavenger in the standard reaction, only a trace amount of **3a** was observed, with a benzylic TEMPO adduct isolated in 43% yield (Fig. 4A). This result suggests that the reaction may be operating through a radical mechanism involving the formation of a benzylic radical. Then we examined the epoxide ring opening process (Fig. 4B). No reaction was observed between epoxide **2a** and NaI in the absence of Ru catalyst or acid additive (entry 1). With the addition of 5% Ru catalyst iodohydrin **7**, formed by iodide opening at the less hindered C−O bond of **2a**, was produced in 7%. Interestingly, iodohydrin **6**, resulting from cleavage of benzylic C−O bond of **2a**, was obtained in only trace amount (entry 2). This result stands in stark contrast with Doyle[68] and Yang's[69] stoichiometric experiments between **2a** and HI as well as NaI, respectively, where **6** was formed as the single product in excellent yields[70]. The addition of acid additive (**A4**) significantly promoted the generation of both iodohydrin **6** and **7**, with **7** still the predominant regioisomer. At the same time, ester substituted products **8** and **9** were also obtained in the reaction, with **9** resulting from the cleavage of the less hindered C−O bond of **2a**, as the major isomer (entry 3). Similar results were obtained when both the Ru catalyst and acid **A4** were simultaneously used (entry 4). Interestingly, in the absence of NaI, acid **A4** could not promote the ring opening of **2a** (entry 5) by itself; However, in the presence of the Ru catalyst, **8** and **9** were obtained in high yield (entry 6). These results reveal that a number of ring opening pathways are available in the reaction conditions, with the major products resulting from the opening of the epoxide at the less hindered C−O bond. Puzzlingly, this regioselectivity is reversed compared to that observed in the alkylation products **3**, which are formed with complete selectivity at the most hindered C−O bond. When **6** was reacted with **1a** under the standard conditions, **3a** was not detected (Fig. 4C). However, upon addition of Na₂CO₃ as a base, **3a** was formed in 70% yield. These results are consistent with an iodide promoted epoxide opening from the benzylic C−O bond with concomitant in situ formation of a base that is required to carry out the C−H activation step. Suprisingly, when **7** was used instead in the above reaction, identical product **3a** was obtained in the presence of Na₂CO₃

(Fig. 4D). Treatment of **6** under the reaction conditions, in the absence of arene, led to formation of regioisomer **7**, as well as both **8** and **9** (Fig. 4E). Similar results were obtained when **7** was used instead of **6** (Fig. 4F). No formation of product **3a** was observed when **8** or **9** were used instead of **6** and **7** in conditions analogous to those in Fig. 4C (Fig. 4G). Taken together, these results indicate a reversible ring opening of **2a** is operating in the reaction. On the other hand, analogous mechanistic studies on the use of oxetanes as coupling partners suggest in that case a direct and non-reversible iodide-mediated regioselective opening of the oxetane may be responsible for the observed *meta*-alkylation (for details, see the Supplementary Discussion section, Mechanistic Studies of Oxetane Involved *meta*-Alkylation Reaction). Finally, when the standard reaction was carried out in the presence of CD₃OD (300 mol %) and stopped before complete conversion, deuterium incorporation (28%) was observed at the *ortho*-position of the recovered starting material $d_5$-**1a**, indicating that the *ortho*-C−H ruthenation is reversible (Fig. 4H). Meanwhile, an intermolecular kinetic isotope effect experiment revealed a relatively small KIE ($k_H/k_D = 1.6$, Fig. 4I), consistent with a non-rate determining reversible C−H cleavage[71].

Based on the above results and literature reports[72] a tentative mechanism is proposed (Fig. 5). With the assistance of Ru catalyst and acid additive, NaI is engaged in nucleophilic ring opening of **2a**, to yield secondary benzylic iodide **V** and primary alkyl iodide **VI**. On the other hand, the aromatic substrate goes *ortho*-cycloruthenation to deliver the cyclometalated Ru-species **I**. Subsequently, **I** goes through single electron transfer with the more reactive intermediate **V** to afford a benzylic radical and a Ru(III) intermediate **VII**, followed by addition of the benzylic radical to intermediate **VII** at the *para*-position to the ruthenium center, to deliver the *meta*-alkylation product **3** and close the catalytic cycle. Meanwhile, based on results presented on Fig. 4B−F, the generation of primary alkyl iodide **VI** should dominate the opening of **2a** in the reaction, however it is an unproductive pathway in the reaction. This is particularly unexpected since cyclometalated-Ru-species **I** is known to undergo facile *ortho*-alkylation with primary alkyl halides[73,74]. The absence of such by-product in our reaction (as well as the result in Fig. 4D−F) suggests that a fast-reversible equilibration of **VI** and **V** via **2a**, as well as a catalyst-controlled discrimination of **VI** and **V** are simultaneously in operation, accounting for the strong regioselectivity obtained in our reaction. To the best of our knowledge, a reversible iodide-mediated epoxide ring opening dynamic kinetic process has not been reported previously in a transition metal-catalyzed coupling reaction.

In conclusion, through a process co-catalyzed by Ru, a carboxylic acid and iodide, we have achieved remote *meta*-alkylation using epoxides as alkyl donors. In addition, unique tandem alkylation/cyclization, the employment of typically challenging substrates such as unsymmetrical 1,2-disubstituted epoxide, complex pharmaceutical compounds and previously unreactive oxetane-based alkylating partners further showcase the broad utility of this transformation. Our mechanistic studies reveal an unexpected catalyst-controlled dynamic kinetic regioselection process is responsible for the high selectivity of the reaction. We envision that this method will encourage the development of more diverse and selective epoxide ring opening reactions.

## Methods

### Representative procedure for Ru-catalyzed *meta*-C−H alkylation with epoxides

In a glove box, an oven-dried crimp-cap microwave vial equipped with a magnetic stirring bar was charged with Ru(PPh₃)₃Cl₂ (5.0 mol %), NaI powder (1.0 equiv) and 2-ethylbutyric acid (30 mol %), then substrates **1** (0.20 mmol), epoxide **2** (2.0 equiv) and dioxane (1.3 M) were added. The vial was then capped and taken out of glovebox, stirred at 70 °C for 44 h. The reaction was then allowed to cool to room temperature and concentrated *in vacuo*. The residue was purified by column chromatography under the conditions noted to yield the desired product.

**A** Radical trapping experiments: **1a** + **2a** (2.0 equiv) → *standard conditions see Scheme 3* TEMPO (2.0 equiv) → **3a**, trace + 43% (isolated)

**B** **2a** (2.0 equiv) + NaI (0.2 mmol), dioxane (1.3 M), 70 °C, 44 h → Results

| Entries | Condition variation | Results [a] |
|---|---|---|
| 1 | As above | no reaction |
| 2 | add 5% Ru(PPh$_3$)$_3$Cl$_2$ | **6**: < 0.5%; **7**: 7% |
| 3 | add 30% 2-ethylbutanoic acid (**A4**) | **6**: 5%;  **7**: 25%  **8**: 7%[b];  **9**: 21%[b] |
| 4 | add 5% Ru(PPh$_3$)$_3$Cl$_2$ with 30 % 2-ethylbutanoic acid (**A4**) | **6**: 5%;  **7**: 24%;  **8**: 10%[b];  **9**: 33%[b] |
| 5 | no NaI 30% 2-ethylbutanoic acid (**A4**) | **8**: 0%;  **9**: 0% |
| 6 | no NaI, 5% Ru(PPh$_3$)$_3$Cl$_2$ 30% 2-ethylbutanoic acid (**A4**) | **8**: 30%[b];  **9**: 70%[b] |

Compounds **6**, **7**, **8**, **9**

**C** **1a** + **6** (2.0 equiv) → *standard conditions [c]* : **3a**, 0%; *standard conditions [c] with Na$_2$CO$_3$ (2.0 equiv)* : **3a**, 70%

**D** **1a** + **7** (2.0 equiv) → *standard conditions [c]* : **3a**, 0%; *standard conditions [c] with Na$_2$CO$_3$ (2.0 equiv)* : **3a**, 25% w/o other isomers

**E** **6** → Ru(PPh$_3$)$_3$Cl$_2$ (5 mol %), 2-ethylbutanoic acid (30 mol %), Na$_2$CO$_3$ (2.0 equiv), dioxane (1.3 M), 70 °C, 44 h → **7** + **8** + **9**;  23%  43%[b]  57%[b]

**F** **7** → Ru(PPh$_3$)$_3$Cl$_2$ (5 mol %), 2-ethylbutanoic acid (30 mol %), Na$_2$CO$_3$ (2.0 equiv), dioxane (1.3 M), 70 °C, 44 h → **6** + **8** + **9**;  3%  20%[b]  47%[b]

**G** **1a** + **8** or **9** (2.0 equiv) → *standard conditions [d]* : **3a**, 0%; *standard conditions [d] with Na$_2$CO$_3$ (2.0 equiv)* : **3a**, 0%

**H** **1a** + **2a** (2.0 equiv) → *standard conditions* CD$_3$OD (3.0 equiv), 8 h → *d*-**1a**, 48% (H 0.28 D)

**I** **1a**/*d$_5$*-**1a'** (1:1) + **2b** (2.0 equiv) → *standard conditions* 8 h, K$_H$ / K$_D$ = 1.6 → **3o**/*d$_4$*-**3o**, 34%

**Fig. 4 | Mechanistic studies. A** Radical trapping experiments. **B** Epoxide opening studies. **C** Testing iodohydrin **6** as the potential reaction intermediate. **D** Testing iodohydrin **7** as the potential reaction intermediate. **E, F** The experiments to prove the epoxide ring opening is a reversible process. **G** Testing ester substituted compounds **8** and **9** as the potential reaction intermediate. **H** Deuteration experiments with CD$_3$OD. **I** Intermolecular kinetic isotope effect experiment. [a] Determined by $^1$HNMR analysis of crude material, and yields were reported based on NaI as the limiting reagent. [b] Yields were reported based on 2-ethylbutanoic acid (**A4**) as the limiting reagent. [c] Without NaI.

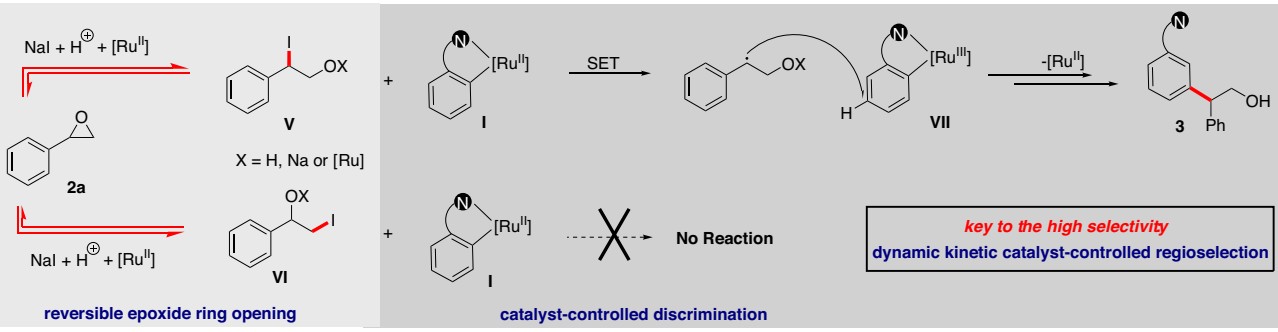

**Fig. 5 | Proposed reaction mechanism.** Proposed catalytic cycle that including a sequence of reversible epoxide ring opening and catalyst-controlled discrimination.

## Data availability

All data generated or analyzed during this study are included in this Article and the Supplementary Information. Details about materials and methods, experimental procedures, mechanistic studies, characterization data, computational details, NMR and HPLC spectra are available in the Supplementary Information. All other data are available from the corresponding author upon request.

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

## Acknowledgements

G.-W. W. thanks the National Natural Science Foundation of China (No. 22201114), the Fundamental Research Funds for the Central Universities (lzujbky-2022-17). We gratefully acknowledge the European Research Council (ERC) for an advanced grant (RuCat, 833337) to I. L. and the Engineering and Physical Sciences Research Council (EPSRC, EP/S02011X/1) for funding to I.L.

## Author contributions

G.-W.W. and I.L. conceived the project. G.-W.W. carried out initial discovery work. G.-W.W., P.-B.B. and A.D. performed optimization of the methodology and exemplification of the chemistry. G.-W.W. and I.L. secured the funding and directed the work. The manuscript was prepared by G.-W.W. and I.L. with contributions from all other authors.

## Competing interests

The authors declare no competing interests.
