## [Peer Review File · Nature Communications]

REVIEWER COMMENTS

Reviewer #1 (Remarks to the Author):

This paper describes the meta-selective alkylation of aromatic rings bearing directing groups using a Ru catalyst and epoxides. Similar meta-alkylation reactions using alkyl halides have been reported by Ackerman and Frost. Mechanistic studies have revealed that the reversible ring-opening of epoxides by NaI generates alkyl iodides, which act as alkylating agents. Therefore, from a mechanistic perspective, it can be considered an extension of the known reactions. However, from a synthetic organic chemistry standpoint, it is noteworthy that epoxides are used as alkylating agents, controlling the meta-selectivity in C-H functionalization and the regioselectivity in the ring-opening reaction. This is certainly an important catalytic reaction in organic synthesis and could be suitable for publication in Nature Communications. One point to consider is whether the proposed reaction mechanism is also applicable to the reaction with oxetanes, and it would be beneficial to provide further verification in this regard.

Reviewer #2 (Remarks to the Author):

The authors report on meta-alkylation of phenylpyridine derivatives with epoxides and oxetanes under ruthenium catalysis conditions. The mechanism which is proposed involves radical processes and implies a dual role of the ruthenium catalyst as redox active species and for directed ortho-CH activation/cyclometalation. This type of mechanism has already been reported to explain the regioselective meta-functionalization of phenylpyridine derivatives, which are classical model substrates for this type of catalysis (see ref. 24 for instance). The novelty lies in the use of epoxides as alkylating reagent, and more interestingly the regioselectivity of the created C-C bond, which is reverse to classical epoxide opening. The mechanism has been studied in details leading to the conclusion that a catalyst-controlled dynamic ring opening/ring closing of the epoxide leading to the production of the reactive radical precursor was operating.

Concerning the scope of the reaction, it is noticeable that no example of phenylpyridine featuring a meta-substituted phenyl ring is reported (only ortho and para). In the same direction, no bis-meta-alkylated products are formed. Could the authors comment on this? Is it steric?

The extension to oxetanes with the same regioselectivities is a good achievement, which represents another innovation of this research.

Concerning the mechanism, it is noted that the role of the generated base (carboxylate) is required for the C-H activation. It should be useful to mention the nature of this base in the text (concomitant in situ formation of a base that is required ...). By the way in the catalytic cycle, both deprotonation

and protonation steps are required, and there is only 30% of carboxylic acid in the catalytic system. Would an additional base (carbonate, carboxylate) bring a positive effect? This has apparently not been investigated in the optimization process (Table 1). Another point is the fate of $\text{RuCl}_2(\text{PPh}_3)_3$ in the presence of carboxylic acid at 80 °C. It is known that $\text{RuCl}_2(\text{PPh}_3)_3$ is a good catalyst in radical processes such as atom transfer radical addition (Kharasch reaction) or polymerization (Macromolecules 1996, 29, 6979). It is also known that $\text{RuCl}(\text{O}_2\text{CR})(\text{PPh}_3)_3$ and $\text{Ru}(\text{O}_2\text{CR})_2(\text{PPh}_3)_2$ can be formed and have catalytic properties due to the presence of their internal base (Nat. Commun 2017, 6, 8140; Dalton Trans 2019, 48, 4625). What are the actual catalytic species in this catalytic reaction?

The work presented in this manuscript is very innovative and could be published with the addition of some comments answering some of the above questions.

Reviewer #3 (Remarks to the Author):

In this manuscript, Larrosa et al. described a meta-C–H bond alkylation of aromatics bearing N-directing groups using (hetero)aromatic epoxides as alkylating agents. It is smart to use NaI as the nucleophilic catalyst, which could be engaged in nucleophilic ring opening of epoxides to yield secondary benzylic iodides. Note that the meta-C–H functionalization with various radical precursors such as alkyl halides by Ru-catalysis is well developed by Ackermann and others. In this method, the in situ generated secondary benzylic iodides are acted as actual radical precursors for further meta-C–H alkylation. Therefore, it suffers from the lack of novelty considering the radical precursors. This manuscript would be a very strong publication in a more specified journal.

Response to reviewers

We are grateful to all reviewers for their time in reviewing our work and their useful suggestions to improve the manuscript. Point-by-point responses follow:

Reviewer #1

This paper describes the meta-selective alkylation of aromatic rings bearing directing groups using a Ru catalyst and epoxides. Similar meta-alkylation reactions using alkyl halides have been reported by Ackerman and Frost. Mechanistic studies have revealed that the reversible ring-opening of epoxides by NaI generates alkyl iodides, which act as alkylating agents. Therefore, from a mechanistic perspective, it can be considered an extension of the known reactions. However, from a synthetic organic chemistry standpoint, it is noteworthy that epoxides are used as alkylating agents, controlling the meta-selectivity in C-H functionalization and the regioselectivity in the ring-opening reaction. This is certainly an important catalytic reaction in organic synthesis and could be suitable for publication in Nature Communications.

Question 1: *One point to consider is whether the proposed reaction mechanism is also applicable to the reaction with oxetanes, and it would be beneficial to provide further verification in this regard.*

Response: To explore the reaction mechanism when using oxetanes we have carried out the mechanistic studies detailed in the scheme below.

In eq A, the reaction between 2-phenylpyridine **1a** and iodohydrin **10** occurs smoothly to yield **5a** in 46% yield, indicating that iodohydrin **10** may serve as the key intermediate for *meta*-alkylation of **1a** and 2-phenyloxetane **4**. This result parallels those obtained with the epoxide. However, while using the iodohydrin **11** under identical conditions, no detectable amount of **5a** is obtained (eq B). Furthermore, iodohydrin **10** and **11** do not interconvert under the reaction conditions (eq C and eq D). Moreover, the ring opening study of **4a** in eq E demonstrates that iodide promoted oxetane opening can only occur at more hindered benzylic C–O bond. These results are in stark contrast to those obtained with epoxides where a dynamic equilibrium was observed. Finally, the reaction does not occur in the absence of NaI (eq F). Taken together, these results imply that iodohydrin **10** is generated as the single regioisomer in the reaction when 2-phenyloxetane reacts with iodide, and then functions as the intermediate in the *meta*-alkylation reaction.

ACTION: We have added the above data and a discussion into the SI (pages S70-S74, mechanistic studies section). We have added the following sentence in the mechanistic section of the manuscript: “On the other hand, analogous mechanistic studies on the use of oxetanes as coupling partners suggest in that case a direct and non-reversible iodide-mediated regioselective opening of the oxetane may be responsible for the observed *meta*-alkylation (see SI for more details).”

Reviewer #2

The authors report on meta-alkylation of phenylpyridine derivatives with epoxides and oxetanes under ruthenium catalysis conditions. The mechanism which is proposed involves radical processes and implies a dual role of the ruthenium catalyst as redox active species and for directed ortho-CH activation/cyclometalation. This type of mechanism has already been reported to explain the regioselective meta-functionalization of phenylpyridine derivatives, which are classical model substrates for this type of catalysis (see ref. 24 for instance). The novelty lies in the use of epoxides as alkylating reagent, and more interestingly the regioselectivity of the created C-C bond, which is reverse to classical epoxide opening. The mechanism has been studied in details leading to the conclusion that a catalyst-controlled dynamic ring opening/ring closing of the epoxide leading to the production of the reactive radical precursor was operating.

Question 1: Concerning the scope of the reaction, it is noticeable that no example of phenylpyridine featuring a meta-substituted phenyl ring is reported (only ortho and para). In the same direction, no bis-meta-alkylated products are formed. Could the authors comment on this? Is it steric?

Response: We have tested four meta-substituted phenylpyridines with two of them giving product, albeit in lower yield than ortho- and para-substituted substrates:

meta-Substituted phenylpyridines tend to be poor substrates in these reactions due to cyclometalation occurring para to the substitution, due to steric hindrance. This selectivity then hinders the alkylation, as alkylation is thought to occur para to the ruthenium. Thus limiting the reactivity and yield for these substrates. This same reasoning also explains the lack of bis-alkylation

observed in these reactions. Both of the substrates that gave low yield contained a fluorine in the *meta*-position, which does not have a large steric impact and can help direct the cyclometalation next to it, leading to the product.

ACTION: We have added the results of **3k** and **3l** to Fig 2A in the manuscript, and pages S26-S27 in the SI for compounds data, and pages S104-S107 in the SI for copies of NMR spectrum. We have included the following comment in the manuscript: “In accordance with previous Ru-catalyzed *meta*-alkylation,⁴⁰⁻⁵³ *meta*-substitution on the arene is not well tolerated as it forces the cyclometalation to occur on the distal *ortho*-position and then blocks reactivity. This reasoning is also why bis-alkylation does not occur in these reactions.”

The extension to oxetanes with the same regioselectivities is a good achievement, which represent another innovation of this research.

Question 2: *Concerning the mechanism, it is noted that the role of the generated base (carboxylate) is required for the C-H activation. It should be useful to mention the nature of this base in the text (concomitant in situ formation of a base that is required ...). By the way in the catalytic cycle, both deprotonation and protonation steps are required, and there is only 30% of carboxylic acid in the catalytic system. Would an additional base (carbonate, carboxylate) bring a positive effect? This has apparently not been investigated in the optimization process (Table 1).*

Response: This was an intriguing suggestion. We have tested the effect of different additional carboxylate and carbonate in the reaction (Scheme R1). Specifically, we added 15% of carboxylate and simultaneously increased the loading of 2-ethylbutanoic acid (**A4**) from 30% to 60% in the reaction (entries 1-5, Table R1), in order to in situ generate the carboxylate of **A4** (30%) and maintain the same concentration of **A4** (30%) present in the standard conditions. While the addition of K₂CO₃, Cs₂CO₃, Li₂CO₃ and MgCO₃ have a minimal effect on the reaction outcome, Na₂CO₃ slightly inhibits the reaction. Moreover, the addition of 30% of carbonates (**A4** was added in 30%, entries 6-10, Table R1) also show a negligible effect on the yield of **3a**.

Entries	Acid (%)	Base (%)	Yield of 3a ^a
1	2-ethylbutanoic acid (60 mol %)	K ₂ CO ₃ (15%)	72%
2	2-ethylbutanoic acid (60 mol %)	Na ₂ CO ₃ (15%)	55%
3	2-ethylbutanoic acid (60 mol %)	Cs ₂ CO ₃ (15%)	75%
4	2-ethylbutanoic acid (60 mol %)	Li ₂ CO ₃ (15%)	71%
5	2-ethylbutanoic acid (60 mol %)	MgCO ₃ (15%)	72%
6	2-ethylbutanoic acid (30 mol %)	NaO ₂ CPh(30%)	69%
7	2-ethylbutanoic acid (30 mol %)	NaOAc(30%)	68%
8	2-ethylbutanoic acid (30 mol %)	KOAc(30%)	76%
9	2-ethylbutanoic acid (30 mol %)	CsOAc(30%)	71%
10	2-ethylbutanoic acid (30 mol %)	Mg(OAc) ₂ (30%)	72%

^a Yield measured by NMR using 1,3,5-trimethoxybenzene as internal standard

Table R1. The effect of additional base

These results demonstrate that 30% of carboxylic acid additive itself is well-suited for the reaction, likely functioning as a proton shuttle in the reaction: it protonates the alkoxide that is generated from the ring opening of epoxides and oxetanes, and the resulting carboxylate then can act as a base to facilitate *ortho*-C–H ruthenation. In this paradigm, only a catalytic amount of carboxylic acid is needed in the reaction. The results observed are consistent with the small KIE and the H/D scrambling observed in the reaction, both of which suggest that the C–H activation is not a rate determining step.

ACTION: We have added this table of results into the SI (page S17, optimization section). We have added following comment in the revised manuscript to explain the possible roles of carboxylic acid additive: “The carboxylic acid additive may function as a proton shuttle in the reaction by protonating the alkoxide generated after the epoxide ring opening, with the resulting carboxylate acting as a base to facilitate *ortho*-C–H ruthenation.”

Question 3: Another point is the fate of RuCl₂(PPh₃)₃ in the presence of carboxylic acid at 80 °C. It is known that RuCl₂(PPh₃)₃ is a good catalyst in radical processes such as atom transfer

radical addition (Kharasch reaction) or polymerization (Macromolecules 1996, 29, 6979). It is also known that $\text{RuCl}(\text{O}_2\text{CR})(\text{PPh}_3)_3$ and $\text{Ru}(\text{O}_2\text{CR})_2(\text{PPh}_3)_2$ can be formed and have catalytic properties due to the presence of their internal base (Nat. Commun 2017, 6, 8140; Dalton Trans 2019, 48, 4625). What are the actual catalytic species in this catalytic reaction?

Response: This is an interesting point as $\text{Ru}(\text{O}_2\text{CR})_2(\text{PPh}_3)_2$ is definitely a possible intermediate in the reaction. After heating $\text{RuCl}_2(\text{PPh}_3)_3$ with the carboxylic acid in dioxane, only PPh_3 and OPPh_3 were observed by ^{31}P NMR analysis. However, upon addition of K_2CO_3 , $\text{Ru}(\text{O}_2\text{CR})_2(\text{PPh}_3)_2$ was also observed by ^{31}P NMR.

A

B

The standard reaction was stopped after 22 h, diluted in dioxane, filtered and submitted for ^{31}P NMR and mass spectral analysis. $\text{Ru}(\text{O}_2\text{CR})_2(\text{PPh}_3)_2$ was not observed by either analysis.

Instead, the ^{31}P NMR spectrum shows two peaks (23.0 and 32.4 ppm) which are consistent with literature values for mono-cyclometalated phenylpyridine-ruthenium species, with one and two triphenylphosphines bound to the ruthenium respectively.¹ The slight difference in the chemical shift can be explained by change of solvent from d_3 -acetonitrile to 1,4-dioxane, which can both coordinate to the ruthenium. There were an extra 2 peaks that could not be assigned at around 42 ppm, but these do not correspond to either $\text{Ru}(\text{O}_2\text{CR})_2(\text{PPh}_3)_2$ or $\text{RuCl}(\text{O}_2\text{CR})(\text{PPh}_3)_3$.^{2, 3}

The ESI-MS of this reaction mixture shows the masses of monocyclometalated-phenylpyridines (both substrate and product) plus one and two triphenylphosphines. This spectrum is consistent with the tentative assignment of the above ^{31}P NMR.

Formation of $\text{Ru}(\text{O}_2\text{CR})_2(\text{PPh}_3)_2$ was attempted, but unfortunately purification of this complex proved difficult and was only able to be obtained as a mixture with PPh_3 . This mixture was shown to be a competent precatalysis for the reaction, with and to a lesser extent without the acid additive.

However, these results do not prove that this species is present in the reaction, either as an on-cycle or off-cycle intermediate.

References:

1. Leyva, Lida et al. Synthesis of cycloruthenated compounds as potential anticancer agents. *Eur. J. Inorg. Chem.* 3055–3066 (2007).
2. Lynam, J. M., Welby, C. E., Whitwood, A. C. Exploitation of a chemically non-innocent acetate ligand in the synthesis and reactivity of ruthenium vinylidene complexes. *Organometallics*, **28**, 1320–1328 (2009).
3. Andersen, R. A., Mainz, V. V. Preparation of RuCH₂PMe₂(PMe₃)₃Cl, Ru(CH₂PMe₂)₂(PMe₃)₂, and Rh₂(CH₂PMe₂)₂(PMe₃)₄ and their reactions with hydrogen. *Organometallics*, **3**, 675–678 (1984).

ACTION: All above new results and discussions have been added into mechanistic studies section of the SI (pages S74-S76).

The work presented in this manuscript is very innovative and could be published with the addition of some comments answering some of the above questions.

Reviewer #3

In this manuscript, Larrosa et al. described a meta-C–H bond alkylation of aromatics bearing N-directing groups using (hetero)aromatic epoxides as alkylating agents. It is smart to use NaI as the nucleophilic catalyst, which could be engaged in nucleophilic ring opening of epoxides to yield secondary benzylic iodides. Note that the meta-C–H functionalization with various radical precursors such as alkyl halides by Ru-catalysis is well developed by Ackermann and others. In this method, the in situ generated secondary benzylic iodides are acted as actual radical precursors for further meta-C–H alkylation. Therefore, it suffers from the lack of novelty considering the radical precursors. This manuscript would be a very strong publication in a more specified journal.

Response: We thank the reviewer for agreeing on the 'smart' design of the reaction and indicating that the work is a very strong publication. We do however disagree with them on the lack of novelty for the following reasons:

1) Despite there are a few examples of radical precursors in the Ru-catalyzed *meta*-alkylation, this is by no means 'well developed' and there are still massive limitations in the field. To date, secondary and tertiary alkyl halides are predominately used as radical precursors to participate or initiate the *meta*-alkylation, other precursors are rarely explored. The search for more convenient coupling partners with additional features such as ready availability, stability and ease of derivatization following the C-H alkylation is still very much ongoing.

2) The use of epoxides as useful alkylating reagent in *ortho*-C-H bond functionalization has been well explored and documented. Our results represent the first process that uses epoxide as the alkylating reagent in *meta*-alkylation, and this is accomplished with full regioselectivity. In addition, 1,2-disubstituted epoxides, which have been shown to be challenging substrates in *ortho*-alkylations, are well tolerated under our protocol.

3) In our mechanistic studies we uncovered that a reversible iodide-mediated epoxide ring opening process as well as a catalyst-controlled dynamic kinetic regioselection were operating (which to the best of our knowledge has not been previously shown in a transition metal-catalyzed coupling reaction) together contributing to the excellent regioselectivity obtained in our reaction.

4) Our strategy has been successfully extended to oxetanes which also lead to *meta*-alkylation. Indeed, using oxetanes in directed C–H alkylation is a challenging task due to their relative lower reactivity.

REVIEWERS' COMMENTS

Reviewer #2 (Remarks to the Author):

The authors have taken into account all the remarks and suggestions of the referees. They have done additional experiments leading to stronger conclusions. As it is now, it deserves publication without further modifications.

I would just note that the primary alkyl iodide (see Fig. 5) is product VI (not VII as mentioned in the text on page 8).

Reviewer #2 (Remarks to the Author):

The authors have taken into account all the remarks and suggestions of the referees. They have done additional experiments leading to stronger conclusions. As it is now, it deserves publication without further modifications.

Question 1: *I would just note that the primary alkyl iodide (see Fig. 5) is product VI (not VII as mentioned in the text on page 8).*

Response: We have corrected the number of alkyl iodide from **VII** to **VI** in the manuscript.

ACTION: We have modified the following sentence "With the assistance of Ru catalyst and acid additive, NaI is engaged in nucleophilic ring opening of **2a**, to yield secondary benzylic iodide **V** and primary alkyl iodide **VII**" to "With the assistance of Ru catalyst and acid additive, NaI is engaged in nucleophilic ring opening of **2a**, to yield secondary benzylic iodide **V** and primary alkyl iodide **VI**."